# The Effectiveness of Dog Population Management: A Systematic Review

**DOI:** 10.3390/ani9121020

**Published:** 2019-11-22

**Authors:** Lauren M. Smith, Sabine Hartmann, Alexandru M. Munteanu, Paolo Dalla Villa, Rupert J. Quinnell, Lisa M. Collins

**Affiliations:** 1Faculty of Biological Sciences, University of Leeds, Leeds LS2 9JT, UK; bslsmi@leeds.ac.uk (L.M.S.); R.J.Quinnell@leeds.ac.uk (R.J.Q.); 2VIER PFOTEN International, 1150 Vienna, Austria; Sabine.Hartmann@vier-pfoten.org (S.H.); alex.munteanu@vier-pfoten.org (A.M.M.); 3Istituto Zooprofilattico Sperimentale dell’Abruzzo e del Molise “G. Caporale”, 64100 Teramo, Italy; p.dallavilla@izs.it

**Keywords:** *Canis familiaris*, free-roaming, stray, population control, catch-neuter-release, sheltering, culling

## Abstract

**Simple Summary:**

Domestic dogs are abundant worldwide—as owned pets, unowned strays, and feral animals. High numbers of free-roaming dogs can be a concern because of the risks they pose to public health, animal welfare, and wildlife. Using a systematic review process, we investigated what the research published to date can tell us about the effectiveness of different dog population management methods. We found that management methods for dog populations have been researched in multiple countries worldwide, using a wide range of indicators to assess method effectiveness. We outline the results and suggest improvements to help guide future dog population management policy.

**Abstract:**

The worldwide population of domestic dogs is estimated at approximately 700 million, with around 75% classified as “free-roaming”. Where free-roaming dogs exist in high densities, there are significant implications for public health, animal welfare, and wildlife. Approaches to manage dog populations include culling, fertility control, and sheltering. Understanding the effectiveness of each of these interventions is important in guiding future dog population management. We present the results of a systematic review of published studies investigating dog population management, to assess: (1) where and when studies were carried out; (2) what population management methods were used; and (3) what was the effect of the method. We evaluated the reporting quality of the published studies for strength of evidence assessment. The systematic review resulted in a corpus of 39 papers from 15 countries, reporting a wide disparity of approaches and measures of effect. We synthesised the management methods and reported effects. Fertility control was most investigated and had the greatest reported effect on dog population size. Reporting quality was low for power calculations (11%), sample size calculations (11%), and the use of control populations (17%). We provide recommendations for future studies to use common metrics and improve reporting quality, study design, and modelling approaches in order to allow better assessment of the true impact of dog population management.

## 1. Introduction

### 1.1. The Dog Population

Dogs (*Canis familiaris*) have a global distribution and an estimated total population size of around 700 million [1]. Dogs have one of two ownership states—either they are owned or they are unowned. The owned population is dependent upon humans for food, water, and shelter and may have one or more owner (e.g., “community dogs”). The owned dog population includes both dogs that are restricted in their movement to a limited area (e.g., within a fenced yard or under human supervision on walks), and those that are free to roam unrestricted, without human supervision [2]. Unowned dogs (often referred to as stray) do not have an owner but may still depend upon humans directly or indirectly for resources such as food, water, and shelter [2]. Similar to owned dogs, the unowned population’s movements may be restricted or unrestricted—dogs housed in shelters have restricted movement, but street-dwelling dogs have unrestricted movement. Unrestricted dogs are particularly important as the lack of restriction allows them to roam freely, to mate, and to reproduce. This group is commonly referred to as the free-roaming dog population and includes unrestricted owned and unrestricted unowned dogs [3].

Approximately 75% of dogs across the world fit into this free-roaming dog category [1]. Free-roaming dog abundance varies greatly between countries, relating to the habitat type (urban/rural) and human population (e.g., density and cultural/social factors) [2]. Although dog densities vary globally, they can exist in high numbers. For example, densities as high as 719 dogs km^-2^ have been estimated in Maharashtra in India [4]. Where free-roaming dogs exist in high densities, they can be considered an issue in terms of public health (e.g., transmission of rabies and other zoonotic pathogens) [5,6,7], the environment (e.g., threatening the survival of wildlife) [8,9,10], and the animals’ welfare state [2,11,12,13]. Population management therefore typically focusses on free-roaming dogs [14] to control the population size and—depending on the approach taken—to improve dog health and welfare and mitigate against public health and environmental problems [15,16].

### 1.2. Reasons for Managing Free-Roaming Dog Populations

#### 1.2.1. The Impact of Free-Roaming Dogs on Public Health

Free-roaming dogs are associated with the transmission of a number of zoonotic pathogens [17,18,19,20,21], dog bite injuries [22,23,24], and road traffic accidents [2,25]. Dogs are responsible for transmitting over 300 zoonoses to humans [26,27]. They are perhaps best known for the role they play in the spread and maintenance of the rabies virus [28]. This virus is responsible for an estimated 60,000 human deaths per year amounting to an annual economic cost of 8.6 billion US dollars [29]. Dogs are a primary reservoir host of this virus and account for 99% of human-rabies transmissions [28]. Other notable zoonoses include *Leishmania infantum*, the causative agent of zoonotic visceral leishmaniasis [30]; *Echinococcus* spp. (*E. granulosus* and *E. multilocularis*) [31], which causes echinococcosis; and *Toxocara canis,* which causes toxocariasis [32].

#### 1.2.2. The Impact of Free-Roaming Dogs on Wild and Domestic Animals

Free-roaming dogs can compromise the conservation of wild animals through a combination of pathogen pollution (i.e., the spread of pathogens to naive hosts) [8,33,34], predation [35], competition [36], and hybridisation [9,37,38]. It is estimated that dogs have played a role in the extinction of 11 vertebrate species and threaten the survival of at least 188 more species [10]. In addition to the negative consequences that canine pathogens have on human health, these pathogens can have an important impact on the conservation of endangered species. For example, rabies spread by domestic dogs can threaten Ethiopian wolf (*Canis simensis*) [8] and African wild dog (*Lycaon pictus*) [33] populations. The taxonomic relatedness of the domestic dog to other canids, such as dingoes (*Canis lupus dingo*) and wolves (*Canis lupus* ssp.), is also of concern when dogs are free-roaming. Inter-species breeding can result in hybridisation and threaten species survival [9,37,39]. In addition, dogs have been responsible for the reduction of other species through predation and competition [1,40,41]. For example, predation by dogs resulted in the decrease of mountain gazelle (*Gazella gazella gazella*) in Israel [42], puda (*Puda puda*) in Chile [43], and kiwi (*Apteryx australis mantelli*) in New Zealand [44].

Domestic dogs can also be responsible for the killing of livestock [45,46,47,48,49]; in particular, small-and medium-bodied livestock such as sheep, goats, and donkeys [45,46,47,48,49]. The loss of livestock contributes to substantial economic losses [45]. For example, in the USA, this amounts to over $620 million US dollars annually [48]. The financial consequences can be particularly problematic in areas of low-income communities [45]. Additionally, the loss of livestock can increase human–wildlife conflict, as predation by dogs is often mistaken for that of other species, such as wolves [45,46,49] or snow leopards [45]. This can hinder local support for conservation programs.

#### 1.2.3. Health and Welfare of Free-Roaming Dogs

Free-roaming dogs can often experience poor health and welfare conditions. In particular, unowned free-roaming dogs may have an inadequate diet and a high prevalence of starvation and dehydration [2,13]. For example, in India, around 49% of free-roaming dogs have an emaciated body condition [50]. Unowned free-roaming dogs can still be dependent on humans for resources, either directly through feeding or indirectly through the provision of food in human waste [51]. The high prevalence of emaciated body condition state in free-roaming dogs that occurs in some areas may be due to poor quantity or quality of food resources and a high disease burden [50,52]. Free-roaming dogs, particularly those that are unowned, lack veterinary care (such as vaccination or antiparasitics) and are therefore more susceptible to disease. High prevalence of skin conditions and ectoparasites have been reported in several populations [50,53]. Canine transmissible venereal tumour disease is also a welfare concern in free-roaming dog populations, with the prevalence estimated at around 1% in dog populations in Africa, Asia, South America, and Central America [54], although the prevalence has been estimated to be as high as 15% in female dogs in some free-roaming populations [55]. Additional health and welfare risks to free-roaming dogs include injury caused by road traffic accidents, abusive treatment by locals [3], and inhumane methods of removal (e.g., poisoning, electrocution, drowning, or carbon monoxide poisoning [56]).

### 1.3. Methods of Managing Dog Populations

#### 1.3.1. Responsible Groups and Motivations for Dog Population Management

Different groups (e.g., researchers, animal welfare organisations, or government agencies) are often responsible for setting up dog population management programs [15]. They manage the population in three main ways: culling, long-term sheltering, and fertility control of free-roaming dogs [14,15]. In addition, programs may include a focus on public education of responsible ownership and taxation of dog ownership. Different countries, as well as different regions within a country, may vary in their objectives for carrying out dog population management programs [14,15], such as: reducing the number of free-roaming dogs; increasing awareness of responsible ownership practices; or improving the health of the free-roaming dog population. These objectives may be underpinned by: dog-centric motives, such as improving dog health and welfare; human-centric motives, such as the control of zoonotic disease [14] and reduced prevalence of dog bite injuries; and wildlife-centric motives, such as reducing the risk to the conservation of other species.

#### 1.3.2. Culling

Historically, culling has been the primary method used to reduce numbers of free-roaming dogs [57]. Culling is the episodic removal and killing of individuals for the purpose of population reduction. The World Health Organisation published guidelines in 1990 discouraging the use of culling and recommending alternative methods (e.g., registration and identification, vaccination, public education, and sterilisation) [58]. Despite these recommendations, many countries continue to use culling as a primary method of population control [15]. Injectable barbiturates are more commonly used in high- and upper-middle-income countries [15], whereas poisoning and shooting are often used in lower-middle- and low-income countries [15]. National law in some countries (e.g., Bulgaria [59], Italy [60], and Kosovo [61])) prohibits the killing of dogs for the purpose of population control.

#### 1.3.3. Sheltering

In some countries, sheltering free-roaming dogs is the most common method of dog population control. Similar to culling, sheltering aims to reduce the free-roaming dog population size by removal of dogs. Ultimately, sheltered dogs may be: (i) euthanised; (ii) adopted; or (iii) permanently stay in the shelter. Shelters are commonplace globally and may be government-run (public shelters), privately-run, or operated by NGOs. The numbers of dogs coming into the shelter are often greater than the number of dogs going out, for example to be rehomed [62,63,64]. This results in either lifelong stays in the shelter or euthanasia [62,63,64]. As national law in some countries prohibits euthanasia of healthy animals, this can lead to long-term sheltering and overcrowding [65]. Moreover, the use of shelters to house dogs is costly and, as such, more commonly employed in high- and upper-middle-income countries [15]. Due to the expense, this method may be unsuitable in lower-middle- and low-income countries [15].

#### 1.3.4. Fertility Control

Fertility control can be achieved through surgical or chemical sterilisation or contraception [66]. Surgical sterilisation through the catch-neuter-release (CNR) of free-roaming dogs is the predominant method of fertility control. This method involves collecting free-roaming dogs and carrying out spay or castration surgery in either a fixed-location or mobile clinic. CNR has been carried out in several countries and states, for example in Italy [67], India [68,69,70], Bangladesh [71], Sri Lanka [72], and Brazil [73]. Surgical sterilisation is generally more socially acceptable than culling. However, in some locations, there can be conflict between locals and the groups/agencies conducting CNR, as some owned free-roaming dogs are caught and neutered against their owner’s wishes [74]. In some communities, owners are against the surgical sterilisation of dogs due to their religious beliefs or the misunderstanding that neutering causes undesired behavioural changes [75,76]. In addition, CNR has associated expense, as it requires skilled staff, clinical facilities, and medicines.

### 1.4. Study Aims

We conducted a systematic review to synthesise the existing evidence of the effectiveness of different dog population management methods. In this review, we describe: (1) where and when the impact of dog population management has been assessed; (2) what management methods have been used; and (3) what effect the management method had on: (i) the dog population size; (ii) dog health and welfare; (iii) public health risk; (iv) public attitude; and (v) risk to wildlife populations. The effectiveness of dog population management depends upon the management intensity (coverage and length of management); therefore, we reported the effects in relation to these criteria wherever possible. In addition, we evaluated the reporting quality of the relevant published studies to allow weighting of evidence for future decision-making.

## 2. Materials and Methods

### 2.1. Search Strategy

We conducted an initial literature search in February 2017, using the following search engines: Web of Science; ProQuest (Applied Social Sciences Index & Abstracts, PAIS Index, Sociological Abstracts, and Worldwide Political Science Abstracts); LILACS; and Google Scholar (results from Google Scholar were limited to the first 50 pages, due to the high volume of returned literature and lack of relevancy). The search used key words relating to dog population management (Appendix A). We carried out a second search using the same search engines, keywords, and eligibility requirements in January 2019 to include any papers published in the interim period.

### 2.2. Eligibility Requirements

A single corpus of all returned literature was compiled across the searches and cleaned of any duplications prior to filtering. Entries were filtered in three stages, based on the relevance to the study aims. These stages involved assessing the paper’s: (1) title; (2) abstract; and (3) full text. At each stage, papers were included or excluded depending on their match to the following inclusion criteria: (i) one of the primary aims of the literature was to assess, describe, investigate, or compare the impact of unowned free-roaming dog population management, in terms of dog population demographics, dog health and welfare, public attitude, or public health risk; (ii) the study design was observational, intervention or modelling; and (iii) was primary literature. Papers were excluded from the review if: (i) they were not a primary research source; (ii) their study design was systematic review, meta-analysis, lab intervention, or case report; or (iii) they assessed, described, or compared only owned dogs that were not free-roaming (i.e., restricted, owned dogs). This was assessed at Stage (1) (title stage) depending on whether the title included the key words (Appendix A) indicating that the paper met the inclusion criteria. At Stages (2) (abstract) and (3) (full text), this was assessed by whether the text met the above-stated inclusion and exclusion criteria. Studies in all languages were considered, although searches were conducted with keywords in English only. There was no restriction on date of publication.

Papers that passed through all three filtering stages were included for review and are referred to as the final corpus. At Stages (1) and (2) of the filtering process, a second reviewer assessed 3% of the papers (Stage (1) = 150 of 4629 papers and Stage (2) = 30 of 923 papers) to check the level of inter-rater inclusion/exclusion agreement. Any papers that were disagreed upon were disputed and a decision reached jointly by both reviewers (details in the Appendix A).

To increase the possibility of capturing all relevant papers, references from papers in the final corpus were screened using the above three-stage filtering process. All references that matched the above inclusion/exclusion criteria were included in the final corpus.

### 2.3. Information Extraction

The following information was extracted from the final corpus: (i) year of publication, country of study, and its economic status (defined by The World Bank 2019 country income classification [77]); (ii) study impact category (dog health and welfare, dog demographics, public attitude, public health, or wildlife), and dog population management method (culling, sheltering, fertility control, or a combination of methods); and (iii) methods, measurements, and study reporting quality. Reporting quality was assessed based on guidelines from Preferred Reporting Items for Systematic Reviews and Meta-Analyses (PRISMA) [78] and Animals in Research: Reporting In Vivo Experiments (ARRIVE) [79]. For the final corpus, we assessed quality based on: study design, reporting of aims/hypotheses, appropriate study outcome (as defined by [16]), and definition of study population. Study populations were classified into: (i) unowned, free-roaming; (ii) owned, free-roaming; (iii) unowned, restricted; (iv) owned restricted; (v) undefined (i.e., the paper did not report which population was under investigation); or a combination of the five categories.

### 2.4. Evaluating Study Design and Reporting Quality

Where appropriate for the study design, we assessed the study and reporting quality based on: the presence/absence of a power calculation, presence/absence of a sample size calculation, inclusion of a control population, accounting for inter-observer reliability, and reporting of baseline characteristics.

## 3. Results

### 3.1. Year of Publishing, Country of Study and Economic Status

The systematic review resulted in an initial (pre-filtered) corpus of 4863 papers, which we reduced following the three-stage filtering process to 36 papers (Figure 1). To ensure key papers were not missed, the references of included papers were reviewed using the same inclusion/exclusion criteria. This resulted in three additional papers and a final corpus of 39 papers. The final corpus comprised 36 peer-reviewed papers and three theses (two Masters of Science and one Masters of Veterinary Medicine). The papers were published between 1977 and 2018, with 82% published between 2008 and 2018.

Most of the studies were carried out or used data from locations within a single country (87%). These were located in 15 different countries across Africa (3%), Asia (39%), Central America (3%), Europe (18%), North America (10%), and South America (15%), in countries that were high income (27%), upper-middle income (38%), lower-middle income (32%), and low income (3%). A high proportion of the studies was conducted in India (26%) (Appendix A). Three studies used data from multiple countries (8%) and two studies did not specify a country (5%).

### 3.2. Dog Population Management Methods and Impacts

#### 3.2.1. Dog Population Management Methods

The management methods studied in the final corpus included: fertility control through neutering and immunocontraceptives (13 papers, 33%); culling (indiscriminate culling and culling of infected dogs: (7 papers, 18%)); sheltering (2 papers, 5%); and taxation (1 paper, 3%) (Table 1). Combinations of methods were also studied: fertility control and sheltering (9 papers, 23%); fertility control and culling (6 papers, 15%); and fertility control and movement restriction (1 paper, 3%) (Table 2). Of the papers that involved fertility control, 79% (23 of 29 papers) controlled the fertility of both male and female dogs (Table 3 and Table 4). Eight papers (21%) directly compared different methods of management: three compared fertility control and culling (8%); three compared fertility control and sheltering (8%); one compared fertility control and movement restriction (3%); and one compared different taxation methods (3%).

#### 3.2.2. Impact Category and Indicators of Effect

Dog population management methods were investigated in terms of the impact they have on: dog health and welfare (6 papers, 15%); dog demographics (13 papers, 33%); public attitude to free-roaming populations (3 papers, 8%); public health (16 papers, 41%); and risk to wildlife populations (1 paper, 3%) (Appendix A). To evaluate these impacts, the final corpus reported 19 different indicators of effect.

The majority of these were different indicators of dog health and welfare, and public health risk, and relatively few different indicators were used to assess dog demographics and public attitude. Considering all the reported indicators, studies used dog population size most frequently to evaluate impact (19 papers, 49%). Considering all management methods and indicators, studies most often evaluated the effect of fertility control and sheltering using dog population size as an indicator (8 papers, 21%).

### 3.3. Quality Evaluation

We assessed the quality of the intervention and observational studies in the final corpus. We split our measures into two categories: those that applied to all papers (including study design, reporting of aims/hypotheses, appropriate outcome studied, and definition of study population), and those that applied to papers depending on their study design (inclusion of power calculation, sample size calculation, control population, inter-observer reliability, and reporting of baseline characteristics).

#### 3.3.1. Study Design and Study Populations

In the final corpus, 33 papers used only one study design and six papers used two different study designs within the paper (Table 2). Papers in the final corpus used observational (i.e., observing dog population management, but not imposing the intervention themselves) (18 papers, 46%), intervention (1 paper, 3%), modelling (14 papers, 36%), a combination of observational study designs (3 papers, 8%), and a combination of observational and modelling study designs (3 papers, 8%). Of the observational study designs, seven papers used a retrospective cohort (33%), six papers used a longitudinal cross-sectional (29%), nine papers used a single time point cross-sectional approach (38%), two papers combined prospective cohort and retrospective cohort (10%), and one paper combined a single time point cross-sectional and retrospective cohort study design (5%). Papers reported various combinations of dog populations, including: free-roaming owned, free-roaming unowned, restricted owned, and shelter dogs (Table 2). Of the various combinations, 36 papers (92%) investigated both free-roaming unowned and free-roaming owned dogs. Twenty-six of these papers (72%) grouped this population as one (e.g., the free-roaming dog population) and did not distinguish between owned and unowned dogs (Appendix A). Two papers did not define their study population (5%).

#### 3.3.2. Study Reporting Quality Indicators

All papers in the final corpus reported their aims, with the majority aiming to understand the impact of dog population management as a primary objective and others describing methods of dog population estimation, model development, and guideline development. All papers in the final corpus used an appropriate outcome to measure the effect of dog population management (as defined by Hiby et al. (2017) [16]). In the observational/intervention studies, 35% of papers did not report the management coverage and 9% did not report the length of management (Table 2). In general, study quality was low in the observational/intervention papers. Only one study used replication (4%), only six studies investigated different groups (26%) and only four included a control population (17%). Reporting was low for both power calculations (11%) (i.e., a calculation to determine statistical power: the probability of correctly rejecting the null hypothesis) and sample size calculations (11%) (i.e., a calculation to determine the minimum sample size required to answer the study question). However, where appropriate, the reporting of inter-observer reliability (71%) and baseline characteristics was high (80%). Appendix A outlines the results of the reporting quality indicators. Reporting quality (RQ) scores are reported in Table 2.

### 3.4. Effects of Management Methods on Impact Categories—Observational and Intervention Studies

The effects of the different methods of dog population management in observational and intervention studies are summarised in Table 3.

#### 3.4.1. Dog Health and Welfare

The impacts of fertility control alone, sheltering alone, and combined fertility control and sheltering were investigated on dog health and welfare in observational studies. No papers in the final corpus investigated the effect of culling or taxation on dog health and welfare.

Fertility control significantly increased body condition score in two of three papers. This was achieved when fertility control was implemented at an unreported coverage level over two years of management ([50] 100% RQ) and when an 80% coverage was applied to the female free-roaming dog population over both seven and 17 years of management ([70] 25% RQ). Fertility control was associated with reduced prevalence of injuries ([70] 25% RQ: 80% female coverage over seven and 17 years) and had few associated post-operative complications (between 5% and 7%, depending on the length of observation) ([82] 40% RQ). Yoak et al. (2014) ([70] 25% RQ) reported that fertility control (at an unreported coverage level over two years of management) had varying effects on the prevalence of pathogens, depending on the type of pathogen. This paper compared the prevalence of various pathogens between areas where varying levels of fertility control had been applied. Whilst fertility control significantly decreased viruses and most bacteria, it significantly increased the prevalence of ectoparasites (e.g., *Rhipicephalus sanguineus*) and *Brucella canis* over the two years of management. Similarly, Totton et al. (2011) ([50] 100% RQ; unreported management coverage over two years of management) found that neutered dogs were 1.7 times more likely to have a visible skin condition compared to intact dogs.

One study investigated the impact of sheltering on the post-adoptive welfare of previously free-roaming dogs. This study found no significant differences in the prevalence of behavioural problems following adoption, using the behavioural indicators “destructiveness”, “hyperattachment to owner”, “fearfulness”, “aggressiveness”, and “excessive barking” ([80] 0% RQ).

One paper in the final corpus investigated the impact of combined fertility control and sheltering on dog health and welfare. Radisavljevic et al. (2017) ([81] 0% RQ) reported that neutering, transport, and housing in a new environment did not have a significant effect on physiological stress measures (Table 3).

#### 3.4.2. Dog Population Demographics

The effects of fertility control and combined fertility control and sheltering on dog population demographics were explored through observational studies. All applied fertility control to both male and female dogs at various intensities (see Table 3) and all reported a reduction in dog population size. Totton et al. (2010) ([69] 20% RQ) described different results between their study areas. These study areas had various levels of fertility control coverage. In three of their five study areas, they observed a decline in the dog population size (*p* < 0.05) (at 62%, 66%, and 67% coverage), in one they found a decreasing trend (*p* > 0.05), and in one study area they saw no effect of fertility control (87% coverage). Although different results were reported for the impact of fertility control and sheltering, one study reported a significant decrease in population size by 34% when fertility control and sheltering was applied at 43% over nine months of management ([93] 20% RQ). It is important to note that this is a particularly short period of management and these initial results may be the immediate effects of sheltering, rather than fertility control.

#### 3.4.3. Public Attitude

The effect of fertility control alone and fertility control and sheltering on public attitude was explored in two papers. Costa et al. (2017) ([97] 80% RQ) reported no effect of fertility control on the public perception towards the effectiveness of different dog population management methods after three years of fertility control at an unspecified level of coverage. Public attitude, in this study, was quantified using a questionnaire with both open and closed questions. Boey (2017) ([94] 20% RQ) described a positive improvement of public attitude towards the presence of free-roaming dogs after fertility control and sheltering campaigns at an unspecified level of coverage and length. This was measured using qualitative data collected in interviews and discussion groups.

#### 3.4.4. Public Health Risk

The effects of culling, fertility control, sheltering, combined fertility control and culling, and combined fertility control and sheltering on public health risk were explored in observational and intervention studies. Two papers in the final corpus investigated the effect of culling on public health risk. Both reported that culling decreased the prevalence of visceral leishmaniasis in dogs over short-term periods, but did not have a significant effect over long-term periods (at an unreported level of coverage over two years of management ([106] 50% RQ), and 8% coverage over 14 months of management ([105] 20% RQ). One study found that culling significantly decreased the prevalence of visceral leishmaniasis in children (decrease in incidence from 12 cases per 1000 people per year to 2 cases per 1000 people per year, at an unreported coverage level over four years of management) ([106]; 50% RQ). Papers were in agreement that fertility control can reduce public health risk, at the investigated management intensities (see Table 3). Fertility control of 65% of females over an unspecified length of management was associated with a significant reduction in human bite cases (a decrease of five bites per month) ([99] 50% RQ). Sheltering at an unspecified level of coverage over 11 years of management was associated with a reduction in *Echinococcus granulosus* prevalence in humans, livestock, and dogs, but significance was not reported ([5] 0% RQ). The combination of fertility control and culling on public health risk was explored in three observational studies. All studies reported a reduction in *Echinococcus granulosus* prevalence in dogs and in livestock at the reported management intensities (see Table 3), but did not report significant effects. There was no effect of this management method at an unspecified level of coverage on the number of people operated on for *Echinococcus granulosus* cysts over eight years of management ([103] 20% RQ). Combined fertility control and sheltering at various management intensities was associated with a decrease in public health risk. Schurer et al. (2015) ([93] 20% RQ) reported a decrease of 43% of dog parasite prevalence after nine months of population management intervention at 43% fertility control and 33% sheltering coverage.

### 3.5. Effects of Management Methods on Impact Categories—Modelling Studies

The effects of the different methods of dog population management in modelling studies are presented in Table 4. The effects of methods that are directly compared within the final corpus papers are summarised in Table 5.

#### 3.5.1. Dog Population Demographics

The effects of culling, fertility control, sheltering, taxation, and combined fertility control and movement restriction on dog population demographics were investigated through modelling studies. All used dog population size as an indicator of effect. Three modelling studies investigated the effect of culling on dog population demographics. All reported that culling decreased dog population size at the intensity modelled (see Table 4 for management coverage and length). Yoak et al.’s (2016) [85] agent-based model simulated that culling would decrease population size by 13% over 20 years [85] at current capture rates, although the intensity required to achieve this reduction is not reported. All papers reported that fertility control reduced population size at the intensity modelled. The effect varied from a minimum decrease in population size of 14% over 20 years to 78% over 20 years, depending on the neutering coverage [89]. Sheltering at the modelled intensity had little or no effect on dog population size (population decrease of 3% in 10 years [90]), or no effect [87,90]). One paper in the final corpus [92] reported that taxation of dog buyers at various intensities decreased the free-roaming dog population size. Three papers [87,89,96] explored the effect of combined movement restriction and sheltering at various modelled intensities, all reported synergistic effects but this varied from a 5% population decrease in 30 years [96] to a 73% decrease in 20 years [89].

When sheltering was directly compared to fertility control, fertility control was more effective at reducing population size [87,90]. For example, Hogasen et al. (2013) [90] modelled that an increase in fertility control by 20–40% per year reduced the free-roaming dog population size by 34%, compared to only a 3% reduction where sheltering was increased by 10% each year. In studies that directly compared the effects of culling to fertility control on dog population size, culling was less effective at reducing the population size. Yoak et al. (2016) [85] reported that fertility control decreased population size by 75%, compared to approximately only 13% with culling when using model simulations with the same capture probability and intensity of intervention.

#### 3.5.2. Public Health Risk

The effects of culling and fertility control on public health risk were investigated in modelling studies. All papers reported that, at various modelled intensities, culling decreased dog rabies prevalence (decreasing trend [100,101,102]) and rabies basic reproductive number (R0) (decreasing trend [91,107,108]). Fertility control at the modelled intensities also decreased public health risk. Fitzpatrick et al. (2016) [98] reported a reduction in the number of human rabies cases, estimating a 92% decrease in five years of model simulation when an intervention coverage between 25% and 50% was modelled. Carroll et al. (2010) [100] reported that fertility control decreased the prevalence of dog rabies. The modelled intensity of fertility control required to eradicate rabies varied from maintaining 100% coverage for one month to maintaining 25% coverage for over two years. Carroll et al. (2010) [100] directly compared culling to fertility control and reported culling to be just as effective at reducing dog rabies prevalence at the modelled intensities. However, when combined with rabies vaccination, fertility control was more effective than culling at eradicating dog rabies [100].

#### 3.5.3. Wildlife

One modelling study investigated the effect of fertility control on disease risk to wild animal populations. Using an agent-based model, Belsare et al. (2015) [109] reported that fertility control (at unspecified intensities) reduced the risk to the Indian fox (*Vulpes bengalensis*) population, using the number of canine distemper spill over events as an indicator.

## 4. Discussion

### 4.1. Limitations in Assessing Dog Population Management

This systematic review synthesises research papers investigating different dog population management methods. We determined: (1) where and when the impact of dog population management has been described in the published literature; (2) what methods were assessed and at what intensity (coverage and length of management); and (3) what effects were reported. Furthermore, we evaluated the reporting quality of the studies. Papers in the final corpus suggest that most dog population management methods were associated with some effect on the impact of interest, and mostly in a favourable direction (such as decreasing public health risk or dog population size). The interpretation of these results and assessment of the effectiveness of dog population management methods is limited due to the following reasons:Few studies used a study design that would allow causation to be determined (such as intervention or certain observational studies), and many lacked an appropriate number of treatment and control groups (Appendix A) and replication (Table 3). This therefore makes it challenging to distinguish between changes to a population that are caused by the management method, to incidental changes caused by other factors (e.g., reduction in population numbers over a few years caused by environmental or human related factors in the study area).Multiple indicators are used to assess the impact of dog population management (Table 1). It is therefore difficult to compare the effect of the same population management method across different studies, and even more challenging to compare different methods across studies. This makes it difficult to carry out a formal synthesis of results, such as a meta-analysis, to report the combined evidence. For example, we found that different papers reporting on the evaluation of different management methods did not use the same measurement of dog health and welfare. In this example, it does not make substantive sense to compare whether an increase in normal body condition scores of 13% (with fertility control) indicates a greater impact on dog health and welfare compared to a decrease in leukocyte counts by 4 (×109 cell/L) (when fertility control and sheltering are combined). This therefore makes it difficult to directly compare effects between methods.Studies often investigated combinations of population management methods, such as fertility control and sheltering, and fertility control and culling. It is difficult to assess the impact of dog population management when methods are not used in isolation. Even where studies investigated one method alone, it is unclear whether other methods of dog population management were in place, such as sheltering or taxation. Culling might also be under-represented, as the method is often not reported due to lack of public acceptance (e.g., ad-hoc poisoning and drowning).To effectively review the results of dog population management intervention, it is important to not only consider what method was applied, but also how the method was implemented. This means in practice that information about the intensity of management and associated costs (logistics, training, and facilities) are required in order to fully appreciate and contextualise the results. Any management method has the potential to be effective if the intensity is large enough. For example, moving 100% of the dog population into shelters every week would be much more effective than to only 15% of the population once a year. It is therefore important to consider: (i) management coverage; (ii) length of management; and (iii) cost of management when assessing the effectiveness of different methods. Many papers in the final corpus did not provided information about the coverage of management and some did not report the length of management (Table 2). Information about the cost of management was rarely provided, apart from where included as a parameter in modelling studies.

### 4.2. Investigated Methods and Reported Effects of Dog Population Management

The results of this systematic review highlight the scale and increasing interest in dog population management, which has been studied globally with an increase in the rate of publications in the last decade (Appendix A). In particular, fertility control was often investigated, this aligns with increasing interest over recent years in the use of fertility control to manage animal populations in general [110]. Although interpretation of results from the final corpus is limited, we can still draw some tentative conclusions about the impact of the different management methods.

Overall, papers reported that fertility control had positive effects on dog health and welfare, including improved body condition score and reduced presence of injuries and some pathogens. However, this method increased skin conditions and prevalence of ectoparasites. The positive effects on body condition and presence of injuries could be explained by the lack of sex hormones caused by fertility control. This results in a reduced desire to seek out mates, as well as reduced sexual competition, which can cause weight gain [111,112] and decrease aggression between individuals, respectively [113]. Additionally, as fertility control methods (such as CNR) are often combined with vaccination and antiparasitic treatment, an improved health condition may be reflected in an improvement in body condition [50]. The negative effects of fertility control on the prevalence of skin conditions could relate to the specific protocols carried out by the different population management programs, such as the conditions the dogs are kept in pre- and post-surgery and the medical treatment provided (such as antiparasitics). It is therefore important that future groups carrying out dog population management through fertility control ensure they take measures to reduce pathogen transmission in clinical facilities.

The impact of different management methods on dog population demographics was measured solely through dog population size, allowing some level of comparison between papers. The comparison is still limited, however, as these effects were measured across different time scales, applied at different rates (e.g., neutering coverages), and to different populations of dogs (e.g., free-roaming owned and unowned or free-roaming unowned). For example, in the observational studies, the impact of fertility control varied from decreases in population size of 12% in 1.5 years to decreases in size of 40% over 12 years. Although all methods decreased population size, fertility control had the greatest effect in both observational studies [68,69,83,84] and modelling studies [69,85,86,87,88,89,90]. Fertility control decreases dog population size by preventing births, therefore allowing a reduction of numbers as natural deaths occur. This is in contrast with culling and sheltering, which reduce the population size through the removal of individuals, either through death or the moving of dogs into a shelter population. When fertility control was combined with other methods, such as movement restriction and sheltering, synergistic effects were reported [89]. By increasing the rate of fertility control and restriction status of dogs, this would both reduce the opportunities for reproduction and therefore potentially reduce the birth rate even greater than if fertility control had been used alone. Culling, by increasing the death rate of a population, may cause a rapid reduction in population numbers [85,88]. The culling method has been criticised as ineffective at reducing populations over longer periods of time [88]. This was supported by modelling studies that directly compared fertility control and culling. These papers found that, although culling resulted in an initial decrease in dog population size (e.g., a five-year period [88]), fertility control was more effective at reducing the population size over longer periods of time (e.g., a 20-year period) [85,88]. This may be because individuals removed through culling are replaced by new individuals through compensatory breeding or migration from other locations [114], therefore rendering the method less effective in the longer term. It is also important to note that there were no empirical studies investigating the impact of culling on population size, all were modelling studies, and therefore have limitations in the inferences that can be made to real dog populations.

Multiple different measures were reported to assess effects on public health risk, again making it difficult to compare methods directly. Culling had decreasing effects on the various indicators of public health risk in both observational [105,106] and modelling studies [91,100,101,102,107,108]. This contradicts previous literature suggesting that culling is ineffective at controlling disease in free-roaming dogs [115,116]. For the measurement “prevalence of visceral leishmaniasis”, culling only decreased prevalence over shorter study periods of up to two years (e.g., up to 69% over 14 months [105]), and had no effect over longer periods [106]. This is potentially due to other mammalian disease reservoirs that would allow continued transmission of *Leishmania infantum* to the remaining dog population [106] or by the number of free-roaming dogs recovering after culling, through immigration or compensatory breeding mechanisms [114]. In addition, culling decreased both dog rabies prevalence [100,101] and the basic reproductive number of rabies [91,107,108] in modelling studies. However, when disease control through vaccination was included in the analysis, all papers in the final corpus reported that culling was not as effective as vaccination alone [91,102,107,108] or combined vaccination and fertility control [100]. The prevalence of *Echinococcus granulosus* in humans, livestock, and dogs decreased where culling was combined with fertility control [103,104]. The two studies reported either large decline [104] in dog prevalence or complete eradication [103]. However, neither study uses experimental design or statistical analysis that would allow inference to the association between the management methods and the effect. All papers in the final corpus agreed that fertility control decreased the public health risk indicators [99,100,117]. The reduction in human bite cases can be linked to a reduction in sex hormones, which can in turn reduce the occurrence of aggression and dog bites [118]. The impact of sterilisation on owned dog aggression has long been debated within the literature, some studies finding a reduction in aggressive behaviour and others finding no effect, or increased aggression (see McKenzie (2010) [119] for review). In terms of free-roaming dogs, Garde et al.’s (2016) [120] behavioural observations in free-roaming dogs found no decrease in aggressive behaviour towards conspecifics or humans. The reduction in human bite cases reported in our findings may instead be due to an overall reduction in the free-roaming dog population size or a reduction in the number of puppies, therefore reducing protective behaviour of adult dogs [99].

### 4.3. Study Quality and Recommendations for Future Work

Good quality reporting of research methods and results is vitally important in understanding the validity and reliability of research findings. Additionally, research conclusions should be supported by appropriate study design and statistical modelling approaches. To improve reporting quality and study design, we suggest the following simple refinements for studies investigating the effectiveness of these dog population management approaches:

#### 4.3.1. Increase Reporting Quality

Reporting guidelines are available for a number of biological areas (see [78,79,121,122,123]). In general, recommendations include reporting specific details about the experimental design, study subjects, statistical analyses, and modelling approaches. In particular, to increase reporting quality in studies investigating the impact of dog population management, we recommend reporting the following:

##### Power and Sample Size Calculations

We found the reporting of power calculations (11%) and sample size calculations (11%) to be low across the published papers. By not reporting this information, the value of the findings and recommendations resulting from the research are limited [79]. Therefore, power and sample size calculations should be clearly reported to increase reporting quality, replicability, and confidence in results.

##### Defined Target Dog Population under Investigation Using Clear Common Terminology

We suggest grouping dogs into: (i) unowned, free-roaming; (ii) owned, free-roaming; (iii) unowned, restricted; and (iv) owned restricted. Reporting which target dog population is under investigation would allow the effects of dog population management to be compared between different studies and between studies in different countries. This is particularly important where the definition of dog ownership might differ, for example in areas where there are community dogs that are loosely owned.

##### Management Intensity and Cost

Papers in the final corpus often did not explicitly state the length, coverage, and cost of the applied management method. As described above, to assess management effectiveness, it is important to report the length and coverage of management (e.g., the number of dogs neutered as a percentage of the total population). Reporting management coverage requires knowledge of the dog population size and this can be achieved through methods such as mark-recapture (see [124] for review of dog population estimation methods). Conducting population estimation requires time, manpower, and expertise in study design and analysis. Dog population management is often carried out by charities and government agencies (Appendix A) and these organisations may lack the financial and logistical power, as well as the expertise to conduct such ecological methods. This might partly explain the lack of reporting of management coverage. It is also important to consider and report the population management history in the study area, as previous management may impact the effectiveness of successive management. This information may also be difficult to access, but should be reported if available.

#### 4.3.2. Improve Experimental and Statistical Modelling Approaches

##### Experimental Approaches

Where possible, researchers should consider their experimental approach and use an intervention (e.g., randomised control trials) or observational (e.g., cohort studies) study design that allows cause and effect to be determined, therefore allowing assessment of the true impact of dog population management. Where appropriate for the study design (i.e., intervention and analytical observational studies), appropriate numbers of groups and replication should be included, such as multiple treatment and control groups. This ensures that any effects reported are caused by the dog population management method and not by other causes (e.g., differences in population numbers between years due to differing mortality rates because of weather or other events).

##### Statistical Modelling Approaches

Due to practical, logistical, and financial constraints, studies that are observational (including cohort, cross-sectional, and case-control studies) are often the only feasible options for assessing dog population management. These studies may result in datasets that include large variability due to biological processes, sampling methods, and context (e.g., the specific country the study was conducted). Statistical modelling approaches can be used to deal with the limitations in inferring causal relationships using observational data. These include controlling for variables in statistical models, or matching of additional variables. We recommend using approaches such as directed acyclic graphs [125,126] to help to identify when controlling for variables in the statistical analysis is appropriate, and what variables to control for (see [127] for primer on creating acyclic graphs, dealing with measurement error, and statistically controlling for variables). For example, in a study investigating the impact of population management on dog health, it would be appropriate to control for age of dogs, as young dogs have a different probability of developing certain health conditions than older dogs. Therefore, if age were not controlled for in these analyses, the causal relationship between population management and dog health might not be observed. Additionally, process based modelling approaches have been developed to incorporate the underlying processes in the statistical analysis [128]. These modelling approaches incorporate both the sampling and biological processes that create the patterns observed in the data (e.g., hidden process models [129]), leading to better interpretation of complex causal relationships, where context creates differing outcomes. An example of datasets where hidden process models could be used includes data collected about dog population size through mark–recapture methods or citizen science [129,130,131]. This would therefore incorporate the processes involved in observing dogs (sampling process—e.g., detection probability) and the biological processes involved in the dogs being in the sampling area (biological process—e.g., the probability of migration, birth, or death). It is worth noting that these approaches require statistical and modelling knowledge that may be challenging for the organisations involved in dog population management to acquire/access.

## 5. Conclusions

Our systematic review found that dog population management is conducted in many countries globally [15,16], carried out by different groups (e.g., researchers and animal welfare or government agencies), applying different methods to different populations types (restricted and unrestricted) and using different indicators to monitor the impact of the intervention. It is therefore difficult to synthesise the evidence base and assess the true impact of dog population management techniques [3,16], despite the quantity of work being conducted. Very few of the reviewed studies allowed robust conclusions to be drawn. We recommend that future studies: (i) increase reporting quality; (ii) clearly define target populations; and (iii) increase the use of study design and modelling approaches that allow causality to be determined, in order that cross-study data synthesis and learning can be conducted for a stronger evidence base to support interventions.

## Figures and Tables

**Figure 1 animals-09-01020-f001:**
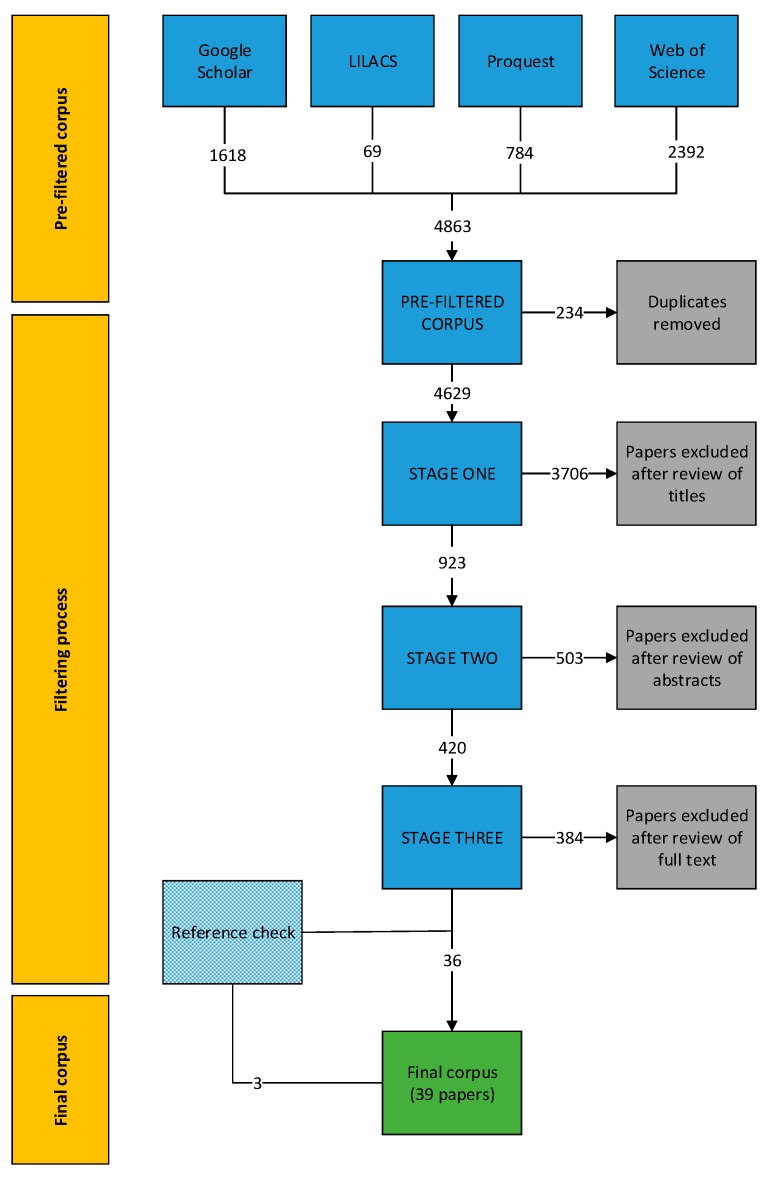
Number of papers included and excluded at each stage of the systematic review process. Grey boxes indicate the number excluded at each stage and the green box indicates the number of papers included in the final corpus.

**Table 1 animals-09-01020-t001:** Impact categories and indicators of effect used in the final corpus to evaluate the effects of management methods. Study design is indicated with either O/I indicating an observational or intervention study, or M for a modelling study. Following the indication of study design is the number of papers (denoted = *n*, where n is the number of papers) adopting this design to test this combination of dog population management method, indicator, and measured impact, followed by the reference details for the relevant papers.

Impact	Indicators	Fertility Control	Culling	Sheltering	Taxation	Fertility Control and Sheltering	Fertility Control and Culling	Fertility Control and Movement Restriction
Dog health and welfare	Body condition score	O/I = 3 [50,70,71]						
	Measure of dog behaviour			O/I = 1 [80]				
	Physiological stress measures					O/I = 1 [81]		
	Presence of injury	O/I = 1 [70]						
	Visible skin condition	O/I = 2 [50,71]						
	Dog disease prevalence (ectoparasites, viruses or bacterial infection)	O/I = 1 [70]						
	Fertility control related complications	O/I = 1 [82]						
Dog population demographics	Dog population size	O/I = 4 [68,69,83,84] M = 7 [69,85,86,87,88,89,90]	M = 3 [85,88,91]	M = 2 [87,90]	M = 1 [92]	O/I = 5 [67,73,93,94,95]		M = 3 [87,89,96]
Public attitude	Public attitude towards free-roaming dogs	O/I = 1 [97]				O/I = 1 [94]		
Public health risk	Number of human rabies cases	O/I = 2 [68,72] M = 1 [98]				O/I = 1 [95]		
	Human bite cases	O/I = 1 [99]						
	Dog rabies prevalence	M= 1 [100]	M = 3 [100,101,102]					
	*Echinococcus granulosus* prevalence in humans			O/I = 1 [5]			O/I = 1 [103]	
	*Echinococcus granulosus* prevalence in livestock			O/I = 1 [5]			O/I = 2 [103,104]	
	*Echinococcus granulosus* prevalence in dogs			O/I = 1 [5]			O/I = 2 [103,104]	
	Dog disease prevalence (visible skin conditions, ectoparasites, viruses or bacterial infection)					O/I = 1 [93]		
	Prevalence of visceral leishmaniasis in dogs		O/I = 2 [105,106]					
	Prevalence of visceral leishmaniasis in children		O/I = 1 [106]					
	Rabies R0		M = 3 [91,107,108]					
Risk to wildlife populations	Canine distemper prevalence in wildlife populations	M = 1 [109]						

**Table 2 animals-09-01020-t002:** All final corpus papers by management factors (method and intensity), study design factors, and reporting quality. Management intensity is reported in terms of coverage and length. Length is reported as years of: (i) mgmt. = management (indicating the study and management method took place at the same time) or (ii) study (indicating the study took place after management began). NA = not applicable for the study design.

Paper	Dog Population Management Method	Management Intensity: Coverage (C) and Length (L) of Management/Study	Dog Population Type	Study Design	No. Replicates	No. Groups	Reporting Quality Indicator Score
[91]	Culling	Up to 33%	Free-roaming stray, Free-roaming owned, Restricted owned	Modelling	NA	NA	NA
[101]	Culling	Various	Free-roaming stray, Free-roaming owned, Restricted owned	Modelling	NA	NA	NA
[102]	Culling	5% and 10%	Undefined	Modelling	NA	NA	NA
[107]	Culling	Various	Free-roaming stray, Free-roaming owned, Restricted owned	Modelling	NA	NA	NA
[108]	Culling	Various	Undefined	Modelling	NA	NA	NA
[106]	Culling	C: Not reported L: first 2 years mgmt	Free-roaming stray, Free-roaming owned	Intervention	1	2 (management and control)	50% (2/4)
[105]	Culling	C: 8% L: 14 months study	Free-roaming stray, Free-roaming owned	Observational-cross-sectional-longitudinal	1	1	20% (1/4)
[86]	Fertility control	Various (65% and above)	Free-roaming stray, Free-roaming owned, Restricted owned	Modelling and Observational-cross-sectional-single time point	NA	NA	NA
[98]	Fertility control	25 to 50%	Free-roaming stray, Free-roaming owned, Restricted owned	Modelling	NA	NA	NA
[109]	Fertility control	Simulate a 50%, 75% and 90% reduction, but do not specify what neutering rate would achieve this	Free-roaming stray, Free-roaming owned	Modelling	NA	NA	NA
[97]	Fertility control	C: NR L: 3 years study	Free-roaming stray, Free-roaming owned, Restricted owned	Observational-cross-sectional-longitudinal	1	1	80% (4/5)
[84]	Fertility control	C: 15% of males and 31% of females L: 1.5 years mgmt	Owned (free-roaming), Owned (restricted)	Observational-cross-sectional-single time point	1	1	50% (1/2)
[82]	Fertility control	C/L: NA	Free-roaming stray, Free-roaming owned	Observational-cohort-prospective and Observational-cohort-retrospective	1	1	40% (2/5)
[99]	Fertility control	C: 65% of females L: Not reported	Free-roaming stray, Free-roaming owned	Observational-cohort-retrospective	1	1	40% (2/5)
[70]	Fertility control	C: ~80% of females L: Various—17, 7, and 0 years mgmt	Free-roaming stray, Free-roaming owned	Observational-cross-sectional-single time point	1	3 (2 CNR intensities and a control)	25% (1/4)
[69]	Fertility control	C: 62 to 87% L: 2 years mg	Free-roaming stray, Free-roaming owned	Observational-cross-sectional-longitudinal and Modelling	6	1	20% (1/5)
[50]	Fertility control	C: Not reported L: 2 years mgmt	Free-roaming stray, Free-roaming owned	Observational-cross-sectional-single time point	1	2 (CNR and control)	100% (3/3)
[83]	Fertility control	C: Not reported L: 12 years study	Free-roaming stray, Free-roaming owned	Observational-cross-sectional-longitudinal	1	1	0% (0/3)
[68]	Fertility control	C: 65% of females L: 8 years mgmt	Free-roaming stray, Free-roaming owned	Observational-cross-sectional-longitudinal and Observational-cohort-retrospective	1	1	0% (0/4)
[71]	Fertility control	C: 61% L: 2 years mgmt	Free-roaming stray, Free-roaming owned	Observational-cross-sectional-single time point	1	2 (CNR and control)	0% (0/1)
[85]	Fertility control and culling	Various	Free-roaming stray, Free-roaming owned	Modelling	NA	NA	NA
[88]	Fertility control and culling	Various	Free-roaming stray	Modelling	NA	NA	NA
[100]	Fertility control and culling	Various	Free-roaming stray, Free-roaming owned	Modelling	NA	NA	NA
[72]	Fertility control and culling	C: Fertility control 3% (max). Culling 10% L: 30 years study	Free-roaming stray, Free-roaming owned	Observational-cohort-retrospective	1	1	25% (1/4)
[103]	Fertility control and culling	C: Not reported L: 8 years mgmt	Free-roaming stray	Observational-cohort-retrospective	1	1	20% (1/4)
[104]	Fertility control and culling	C: Fertility control: 8%. Culling: 67% L: 4 years mgmt	Free-roaming stray	Observational-cross-sectional-longitudinal	1	1	20% (1/4)
[89]	Fertility control and movement restriction	Various	Free-roaming owned	Modelling and Observational-cross-sectional-single time point	NA	NA	NA
[87]	Fertility control and sheltering	Various	Free-roaming stray, Free-roaming owned, Restricted owned, Shelter dogs	Modelling	NA	NA	NA
[90]	Fertility control and sheltering	CNR: 20–40% more captures. Sheltering: 10% increase.	Free-roaming stray, Free-roaming owned, Restricted owned, Shelter dogs	Modelling	NA	NA	NA
[96]	Fertility control and sheltering	Various (from 0 up to 0.2 per year)	Free-roaming stray, Free-roaming owned	Modelling	NA	NA	NA
[73]	Fertility control and sheltering	C: 88% L: 14 months study	Free-roaming stray, Free-roaming owned	Observational-cross-sectional-longitudinal	1	2 (management and control)	67% (2/3)
[94]	Fertility control and sheltering	C/L: Not reported	Free-roaming stray, Free-roaming owned	Observational-cross-sectional-single time point	1	1	20% (1/4)
[93]	Fertility control and sheltering	C: Fertility control: 43%. Sheltered: 33% L: 9 months mgmt	Free-roaming stray, Free-roaming owned, Restricted owned	Observational-cohort-prospective and Observational-cohort-retrospective	1	1	20% (1/4)
[67]	Fertility control and sheltering	C: Not reported L: 13 years mgmt	Free-roaming stray, Free-roaming owned	Observational-cohort-retrospective	1	1	0% (0/3)
[95]	Fertility control and sheltering	C: Fertility control: between 0.03 to 12%. Sheltering: NR L: 5 years study	Free-roaming stray, Free-roaming owned	Observational-cohort-retrospective	1	1	0% (0/2)
[81]	Fertility control and sheltering	C/L: NA	Free-roaming stray	Observational-cross-sectional-single time point	1	1	0% (0/3)
[80]	Sheltering	C/L: NA	Free-roaming stray	Observational-cross-sectional-single time point and Observational-cohort-retrospective	1	2 (previously unowned free-roaming; previously owned)	0% (0/3)
[5]	Sheltering	C: Not reported L: 11 years mgmt	Free-roaming stray, Shelter dogs	Observational-cohort-retrospective	1	1	0% (0/4)
[92]	Taxation	NA	Free-roaming stray, Free-roaming owned, Restricted owned, Shelter dogs	Modelling	NA	NA	NA

**Table 3 animals-09-01020-t003:** Results from papers in the final corpus (excluding modelling studies) of the effects of methods of dog population management on the indicators of impact and impact categories. ↑ indicates an increasing effect, ↓ a decreasing effect and n.e. no effect; combinations of different symbols indicate where evidence is conflicting. Where *p*-values were reported, this is included (e.g., *p* < 0.05), NR = *p*-value was not reported, NS = *p*-value not significant. NA = not applicable for the study design. The size of effect is extracted from papers and reported in terms of the years of: (i) mgmt. = management (indicating the study and management method took place at the same time) or (ii) study (indicating the study took place after management began). Where fertility control is included in the dog population management method, (M&F) indicates fertility control was applied to both males and females, (F) indicates only female fertility was controlled. Supporting evidence is provided in references.

Impact Category	Dog Population Management Method	Indicator	Effect	Country of Study	Management Intensity: Coverage (C) and Length (L) of Management	Size of Effect and Confidence Interval (CI)/Error Estimate (EE) Where Reported	Sample Size
Dog health and welfare	Fertility control	Body condition score (1–5 scale)	↑	India	C: Not reported L: 2 years mgmt	[50] (M&F) Normal body condition 1.7 (CI 1.1–2.5) times more likely in sterilised dogs (does not overlap null value, no *p*-value given). Analytical method: logistic regression models and likelihood ratio test.	888 total (439 CNR; 448 control)
					C: ~80% of females L: Various—17, 7, and 0 years mgmt	^c^ [70] (M&F) Normal body condition 13% (No CI) increase in prevalence in high management areas. (Reported significant, values not given). Analytical method: pairwise comparisons.	240 total (106 high intensity; 82 medium intensity; 101 no previous CNR)
			↓	Bangladesh	C: 61% L: 2 years mgmt	^a^ [71] (M&F) Normal body condition 3% decrease in prevalence (NR).	6341
		Fertility control related complications	n.e.	India	C/L: NA	[82] (M&F) Incidence at: 24 h monitoring major complications 3% (2.1–3.6%); minor complications 3% (1.9–3.4%); 4-day monitoring major complications 7% (3.9–11.5%); minor complications 6% (2.8–9.6%) (NR).	2398 (2198 24 h monitoring, 200 4 day monitoring)
		Presence of injury	↓	India	C: ~80% of females L: Various—17, 7, and 0 years mgmt	^c^ [70] (M&F) Decrease of 22% (No CI) in high management areas. (Reported significant, values not given).	240 total (106 high intensity; 82 medium intensity; 101 no previous CNR)
		Prevalence of pathogens (ectoparasites, virus and bacterial infection)	↑↓	India	C: ~80% of females L: Various—17, 7, and 0 years mgmt	^c^ [70] (M&F) Canine parvovirus ↓ 6%, Canine distemper virus ↓ 9%, fleas ↓ 21%, *Ehrlichia canis* ↓32%, *Leptospira serovars* ↓28%, Infectious canine hepatitis ↓ 23%, *Brucella canis* ↑ 7% in high management areas. (Reported significant, values not given). ^c^ ↑ ticks > 28% (high and low fertility control *p* = 0.0001, high and intermediate fertility control *p* = 0.131) (No CIs). Analytical method: Pairwise comparisons.	
		Prevalence of visible skin conditions	↑	India	C: Not reported L: 2 years mgmt	[50] (M&F) ↑ 1.7 (CI 1.3–2.2) times more likely in sterilised dogs (*p* < 0.001). Analytical method: Logistic regression models and likelihood ratio test.	888 total (439 CNR; 448 control)
			↓	Bangladesh	C: 61% L: 2 years mgmt	[71] (M&F) ↓5% (NR).	6341
	Fertility control and sheltering	Physiological stress measures	↓ n.e.	Serbia	C/L: NA	[81] (F) I = immediately after transport; 24 h = 24 h after housing): n.e. Cortisol, Cholesterol, Triglycerides, and lymphocyte. ↓ Glucose < 0.9(mmol/l) (*p* < 0.001) I = 4.5(+/−1.0) to 24 h = 3.6(+/−1.0), ↓ Leukocyte 4(×109 cells/L) (*p* < 0.01) = 15.1(+/−5.9) to 24 h = 11.1(+/−4.8), ↓ Neutrophil 4.2(×109 cells/L) (*p* < 0.001) I = 11.8(+/−4.8) to 24 h = 7.6(+/−3.2) ↓ Leukocyte/neutrophil ratio (*p* < 0.01) I = 7.4(+/−4.2) to 24 h = 4.9(+/−2.5). Analytical method: Non-parametric Mann-Whitney U test.	40
	Sheltering	Prevalence of behavioural problems	n.e.	Turkey	C/L: NA	[80] n.e. Destructive behaviour, hyper-attachment to owner, barking, aggressiveness, fearfulness, and escaping (No CI) (NS). Analytical method: Chi-squared.	75 total (40 previously unowned free-roaming; 35 previously owned)
Dog population demograph-ics	Fertility control	Dog population size	↓	India	C: Not reported L: 12 years study	[83] (M&F) ↓ ~40% ^b^ (NR).	NA
					C: 65% of females L: 8 years mgmt	[68] (M&F) ↓ 28% (NR).	NA
				Brazil	C: 15% of males and 31% of females L: 1.5 years mgmt	[84] (M&F) ↓12% (NR).	NA
			↓ n.e.	India	C: 62 to 87% L: 2 years mgmt	[69] (M&F) Both ↓ n.e. Decrease between 3% (*p* > 0.05) and 51% (*p* < 0.05). Analytical method: Not reported.	NA
	Fertility control and sheltering	Dog population size	n.e.	Italy	C: Not reported L: 13 years mgmt	[67] (M&F) No effect (NR).	NA
				Brazil	C: 88% L: 14 months study	[73] (M&F) No effect (NR). Control (area A): from 81 (66–97) to 94 (75–113). Intervention (area B): from 70 (57–84) to 81 (65–96). Analytical method: Jolly-Seber mark-recapture model.	NA
			↓	Canada	C: Fertility control: 43%. Sheltered: 32% L: 9 months mgmt	[93] (M&F) ↓ 34% (*p* < 0.001). Analytical method: Not reported.	NA
					C/L: Not reported	[94] (M&F) no quantitative data.	18
				Thailand	C: Fertility control: between 0.03 to 12%. Sheltering: NR L: 5 years study	[95] (M&F) ↓ 23% (NR).	NA
Public attitude	Fertility control	Public attitude towards perception of dog management method	n.e.	Brazil	C: NR L: 3 years study	[97] (M&F) n.e. (*p* = 0.774) (No CI). Analytical method: Chi-squared.	354 Pre-management; 70 post-management
	Fertility control and sheltering	Public attitude towards free-roaming dogs	↓	Canada	C/L: Not reported	[94] (M&F) No quantitative data.	18
Public health risk	Culling	Prevalence of visceral leishmaniasis in dogs	↓	Brazil	C: 8% L: 14 months study	[105] ↓ Between 66% and 69% (NR).	328
					C: Not reported L: first 2 years mgmt	[106] Short term: Initial decrease of ↓ 26% (*p* < 0.001). Analytical method: Chi-squared (temporal changes within areas (intervention and control), and Poisson regression for between intervention and control.	Intervention area: 1989–1990 = 235; 1990–1991 = 248; 1991–1992 = 70; 1992–1993 = 131; and 1993 = 164. Control area = not reported.
			n.e.	Brazil	C: Not reported L: 4 years mgmt	[106] Long term: incidence not significantly different between intervention and control (*p* = 0.07). Analytical method: As above.	
		Prevalence of visceral leishmaniasis in children	↓	Brazil	C: Not reported. L: 4 years mgmt	[106] ↓ incidence from 12 cases/1000 inhabitants/year to 2 cases/1000 inhabitants/year (*p* < 0.01). Analytical method: As above.	NA
	Fertility control	Human bite cases	↓	India	C: 65% of females L: Not reported	^b^ [99] (F) ↓ 5 bites per month (*p* < 0.001) ^b^. Analytical method: Linear least squares regression.	NA
		Number of human rabies cases	↓	India	C: 65% of females. L: 10 years mgmt	[68] (M&F) ↓ 100% (NR).	NA
	Fertility control and culling	Number of human rabies cases	↓	Sri Lanka	C: Fertility control 3% (max). Culling 10% L: 30 years study	[72] (M&F) ↓ 82% (NR).	NA
		*Echinococcus granulosus* prevalence in humans	n.e.	Cyprus	C: Not reported L: 8 years mgmt	[103] (F) n.e. on the number of people operated on for *Echinococcus granulosus* cysts (NR).	NA
		*Echinococcus granulosus* prevalence in livestock	↓	Cyprus	C: Not reported L: 5 years mgmt	[103] (F) ↓ overall infection rate (cattle from 0.09% to 0.01%, sheep from 0.03% to 0.02%, and goats from 0.01% to 0.003%) (NR).	1,899,040 total (104,134 cattle; 885,618 sheep; and 909,288 goats)
					C: Fertility control: 8%. Culling: 67% L: 4 years mgmt	[104] (F) ↓ prevalence between 47% to 2% (depending on species and age) (NR).	Not reported
		*Echinococcus granulosus* prevalence in dogs	↓	Cyprus	C: Not reported L: 6 years mgmt	[103] (F) ↓ 100% in dogs (NR).	2391
					C: Fertility control: 8%. Culling: 67% L: 4 years mgmt	[104] (F) ↓ 80% in dogs (NR).	12,213 in 1972; 3947 in 1976
	Fertility control and sheltering	Dog disease prevalence (helminths, *Isospora, Sarcocystis*, *Giardia*, *Cryptosporidium*, *Taenia, Echinococcus* spp, *Dirofilaria immitis, Ehrlichia canis, Borrelia burgdorferi and Anaplasma phagocytophilum*, and *Toxoplasma gondii*)	n.e.	Canada	C: Fertility control: 43%. Sheltered: 33%. L: 9 month mgmt	[93] (M&F) Overall ↓ 43% (*p* < 0.001). Analytical method: Chi-squared.	145 Pre-clinic; 95 post-clinic
		Number of human rabies cases	↓	Thailand	C: Fertility control: between 0.03 to 12%. Sheltering: NR. L: 6 years study	[95] ↓ 15% (NR).	NA
	Sheltering	*Echinococcus granulosus* prevalence in humans	↓	Spain	C: Not reported L: 11 years mgmt	[5] ↓ 97% (NR).	NA
		*Echinococcus granulosus* prevalence in livestock	↓	Spain	C: Not reported L: 11 years mgmt	[5] ↓ 75% (NR).	376 in 1992; 1172 in 1999
		*Echinococcus granulosus* prevalence in dogs	↓	Spain	C: Not reported L: 11 years mgmt	[5] ↓ 79% (NR).	553 in 1989; 1040 in 1998

^a^ Contradictory result within paper, contacted author to confirm correct results. ^b^ Estimated by approximating numbers from figures in paper. ^c^ Alpha value for pairwise post-hoc adjusted to 0.005 to control for multiple comparisons.

**Table 4 animals-09-01020-t004:** Results from only modelling papers from the final corpus of the effects of methods of dog population management on the indicators of impact and impact categories. ↑ indicates an increasing effect, ↓ a decreasing effect, and n.e. no effect; combinations of different symbols indicate where evidence is conflicting. The size of effect is extracted from papers and reported in terms of the years of modelling simulation. Supporting evidence is provided in references.

Impact Category	Dog Population Management Method	Indicator	Effect	Country of Study	Management Coverage	Size of Effect
Dog population demographics	Culling	Dog population size	↓	No specific country	Up to 33%	[91] Decreasing trend.
				North America	Various	[88] Decreasing trend.
				India	Various	[85] * ↓ 13% over 20 years.
	Fertility control	Dog population size	↓	India	62 to 87%	[69] ↓ 69% (80% neutering coverage) over 20 years.
					Various	[85] * ↓ Between 55% and 75% over 20 years.
				Brazil	Various (65% and above)	[86] Decreasing trend.
				North America	Various	[87,88] Decreasing trend.
				Mexico	Various	[89] ↓ Between 14% and 78% (depending on neutering effort and targeting young vs. mixed age dogs) over 20 years.
				Italy	20–40% more captures.	[90] ↓ 34% over 10 years.
			n.e.	India	62 to 87%	[69] n.e. (31% neutering coverage) over 20 years.
	Sheltering	Dog population size	n.e.	North America	Various	[87] n.e. over 30+ years
			↓ n.e.	Italy	10% increase	[90] ↓ 3% and n.e. over 10 years.
	Taxation	Dog population size	↓	No specific country	NA	[92] Decreasing trend.
	Fertility control and movement restriction	Dog population size	↓	Mexico	Various	[89] Between <18% and 73% (depending on neutering effort and confinement level) over 20 years.
				Brazil	Various (from 0 up to 0.2 per year)	[96]: ↓ 5% in 30 years.
				North America	Various	[87]: Decreasing trend.
Public health risk	Culling	Dog rabies prevalence	↓	Parameters from multiple countries	Various	[100,101] Decreasing trend.
				Chad	5% and 10%	[102] Decreasing trend
		Rabies basic reproductive number (R0)	↓	China	Various	[107,108] Decreasing trend.
				No specific country	Up to 33%	[91] Decreasing trend
	Fertility control	Number of human rabies cases	↓	India	25 to 50%	[98] ↓ 92% in 5 years.
		Dog rabies prevalence	↓	Multiple countries	Various	[100] Decreasing trend.
Wildlife	Fertility control	Prevalence of canine distemper in Indian foxes (*Vulpes bengalensis*)	↓	India	Simulate a 50%, 75% and 90% reduction, but do not specify what neutering rate would achieve this	[109] ↓ Between 3 fewer canine distemper spill over events per 10 years (at 50% population reduction) to 6 fewer canine distemper spill over events per 10 years (at 90% population reduction)

* Estimated by approximating numbers from figures in paper.

**Table 5 animals-09-01020-t005:** Summary of methods directly compared within papers. All papers included in the final corpus directly comparing different methods of dog population management used a modelling study design.

Methods Being Compared	Indicator	Effect		Evidence	Most Effective Method
Fertility control and culling		Fertility control	Culling		
	Dog population size	↓	↓	North America	[88] Over a shorter period (5 years), culling was a more effective strategy. Over a longer period (20 years), both methods had similar effectiveness.
				India	[85] Fertility control was more effective than culling, fertility control reduced population size by over 75%, compared to ~13% with culling over 20 years.
	Dog rabies prevalence	↓	↓	Multiple countries	[100] Culling was as effective as fertility control combined with rabies vaccination.
Fertility control and sheltering		Fertility control	Sheltering		
	Dog population size	↓	↓	Multiple countries	[96] Fertility control and adoption, through sheltering, had synergistic effects. Adoption, through sheltering, was the most effective method when comparing the two.
				North America	[87] Fertility control was the most effective, although adoption, through sheltering, worked well in combination with fertility control.
		↓	↓	Italy	[90] Fertility control was the most effective, reducing dog population size by 34%, compared to only 3% in sheltering.
Fertility control and movement restriction		Fertility control	Movement restriction		
	Dog population size	↓	↓	Mexico	[89] Varying size of effect relating to neutering coverage, age of dog neutering and confinement level. Fertility control of owned dogs and dog movement restriction were most effective when used together.
Different taxation methods		Taxation of dog purchases	Subsidy of dog adoption		
	Dog population size	↓	↓	No specific country	[92] Taxation of dog buyers is the most effective option at reducing the number of free-roaming dogs.

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
