# Peer review of "The Effectiveness of Dog Population Management: A Systematic Review"

_animals, 2019, doi:10.3390/ani9121020_

Round 1

Reviewer 1 Report

I couldn't easily figure out how to submit my comments on the revised
manuscript online, so I'm providing them here. Overall, the authors did
a nice job of responding to my comments, and the additional material
makes for a better and more useful paper. I can therefore recommend you
accept the manuscript. I have a few additional comments which are as
follows:

Line 62: I would use the word “estimated” rather than “reported”

Line 114: The high prevalence of emaciation occurs in some areas, but
it is not universal as this sentence implies. I have worked in places
where free roaming dogs that are emaciated or visibly in poor condition
are a small minority of all dogs, and other places where they are very
common.

Line 144: Although obvious to us, it would probably be appropriate to
briefly define what we mean by culling (i.e. episodic removal and
killing of some substantial fraction of the existing dog population)

Line 188, regarding the new sentence: Because many reviewed papers did
not actually involve conducting the management, it’s fine to add
“wherever possible” to the end of this sentence.

Line 690: I appreciate the additions here, but I think this paragraph is
better with the first sentence omitted. Doing so makes is clearer that
when it’s possible to do a more formal, traditional intervention
experiment (section a), that’s great, but when not (which will be most
of the time) there are other approaches to analyze data. I think I might
also change the heading of section a. to “experimental approaches” or
“intervention approaches”, or something like that, instead of “study
design”. Study design is more universal, and applies not just to formal
intervention experiments, but also to the sorts of studies one might
analyze using the techniques outlined in section b.

Line 170: I suspect a statistician who is familiar with these
techniques would choose a different introductory definition for “process
based” models, since the motivation for developing them wasn’t so much
“messy” datasets as it was trying to capture and characterize the
underlying PROCESSES that are most likely to be responsible for
generating a given data set (messy or otherwise), in contrast to testing
some hypothesis about the end-state of a system. Because an
understanding of processes allows the incorporation of various
covariates in flexible ways, it can lead to a more satisfying
interpretation in situations when there is complex causality and
contextually varying outcomes.

Author Response

Thank you again for providing comments. We think this has further improved our manuscript. Please find below responses to your comments.

Comments and Suggestions for Authors

I couldn't easily figure out how to submit my comments on the revised
manuscript online, so I'm providing them here. Overall, the authors did
a nice job of responding to my comments, and the additional material
makes for a better and more useful paper. I can therefore recommend you
accept the manuscript. I have a few additional comments which are as
follows:

Line 62: I would use the word “estimated” rather than “reported”

Amended in text

Line 114: The high prevalence of emaciation occurs in some areas, but
it is not universal as this sentence implies. I have worked in places
where free roaming dogs that are emaciated or visibly in poor condition
are a small minority of all dogs, and other places where they are very
common.

We have amended in text to: “The high prevalence of emaciated body condition state in free-roaming dogs that occurs in some areas may be due to poor quantity or quality of food resources and a high disease burden [50,52].”

Line 144: Although obvious to us, it would probably be appropriate to
briefly define what we mean by culling (i.e. episodic removal and
killing of some substantial fraction of the existing dog population)

Amended in text to: “Culling is the episodic removal and killing of individuals for the purpose of population reduction.”

Line 188, regarding the new sentence: Because many reviewed papers did
not actually involve conducting the management, it’s fine to add
“wherever possible” to the end of this sentence.

Added to the end of sentence.

Line 690: I appreciate the additions here, but I think this paragraph is
better with the first sentence omitted. Doing so makes is clearer that
when it’s possible to do a more formal, traditional intervention
experiment (section a), that’s great, but when not (which will be most
of the time) there are other approaches to analyze data. I think I might
also change the heading of section a. to “experimental approaches” or
“intervention approaches”, or something like that, instead of “study
design”. Study design is more universal, and applies not just to formal
intervention experiments, but also to the sorts of studies one might
analyze using the techniques outlined in section b.

Thank you for your comment, we have taken out the first and second sentence to make this paragraph clearer. We have also changed 2.a. heading to Experimental approaches (instead of study design).

Line 170: I suspect a statistician who is familiar with these
techniques would choose a different introductory definition for “process
based” models, since the motivation for developing them wasn’t so much
“messy” datasets as it was trying to capture and characterize the
underlying PROCESSES that are most likely to be responsible for
generating a given data set (messy or otherwise), in contrast to testing
some hypothesis about the end-state of a system. Because an
understanding of processes allows the incorporation of various
covariates in flexible ways, it can lead to a more satisfying
interpretation in situations when there is complex causality and
contextually varying outcomes.

We have amended this paragraph to be clearer. We have pasted the whole paragraph here for ease:

Due to practical, logistical, and financial constraints, studies that are observational (including cohort, cross-sectional, and case-control studies) are often the only feasible options for assessing dog population management. These studies may result in datasets that include large variability due to biological processes, sampling methods, and context (e.g. the specific country the study was conducted). Statistical modelling approaches can be used to deal with the limitations in inferring causal relationships using observational data. These include controlling for variables in statistical models, or matching of additional variables. We recommend using approaches such as directed acyclic graphs [127,128] to help to identify when controlling for variables in the statistical analysis is appropriate, and what variables to control for (see [129] for primer on creating acyclic graphs, dealing with measurement error, and statistically controlling for variables). For example, in a study investigating the impact of population management on dog health, it would be appropriate to control for age of dogs, as young dogs have a different probability of developing certain health conditions than older dogs. Therefore, if age was not controlled for in this analysis, the causal relationship between population management and dog health might not be observed. Additionally, process based modelling approaches have been developed to incorporate the underlying processes in the statistical analysis [130]. These modelling approaches incorporate both the sampling and biological processes that create the patterns observed in the data (e.g. hidden process models [131]), leading to better interpretation of complex causal relationships, where context creates differing outcomes. An example of datasets where hidden process models could be used includes data collected about dog population size through mark-recapture methods or citizen science [131–133]. This would therefore incorporate the processes involved in observing dogs (sampling process – e.g. detection probability) and the biological processes involved in the dogs being in the sampling area (biological process – e.g. the probability of migration, birth, or death). It is worth noting that these approaches require statistical and modelling knowledge that may be challenging for the organisations involved in dog population management to acquire/access.

Reviewer 2 Report

Nice work on the revision.  A few minor corrections and questions.

Line 334-6: line 334 says 19 papers as most investigated populations and then the next line list 36 papers?

Line 355: this is Table 2 not 3.

Table 2, some obvious order for the papers should be used.  By management type and then by quality?

Author Response

Thank you again for providing comments. We think this has further improved our manuscript. Please find below responses to your comments.

Comments and Suggestions for Authors

Nice work on the revision.  A few minor corrections and questions.

Line 334-6: line 334 says 19 papers as most investigated populations and then the next line list 36 papers?

We now realise this is confusing –the 36 papers include both those populations investigated together (combined as the free-roaming dog population, including owned and unowned) and any study that reported these populations within any other combination (e.g. free-roaming and shelter dogs). We have amended to exclude the sentence “The most investigated population was the combination of free-roaming owned and free-roaming unowned (19 papers, 49%)”, as reporting this result creates confusion.

Line 355: this is Table 2 not 3.

Yes that is correct – thank you, we have now amended.

Table 2, some obvious order for the papers should be used.  By management type and then by quality?

We have now amended in order of management type and quality.

This manuscript is a resubmission of an earlier submission. The following is a list of the peer review reports and author responses from that submission.

Round 1

Reviewer 1 Report

General Comments:

This is a worthwhile compendium of the limited existing information about outcomes associated with dog population management, but given the very limited amount of primary information and its many flaws, I think the premise of the paper needs to be re-stated as described in the next item. As it stands, the paper states a goal of reviewing “the evidence of the effectiveness of different dog population management methods” (line 181) and then repeatedly notes in the Discussion that “it is difficult to compare methods” due to lack of standardization or other flaws in the primary studies and reports.  Stepping back for a moment, the effect of any management attempt is not simply determined by management method, but is also to a large extent a function of the application parameters of that method and scaling.  For instance, culling 100% of dogs present in the management area every week (if that were possible) would be incredibly effective at reducing population size, but culling 40% of dogs every two years (same method, but different implementation) is only temporarily effective at best, and likely to be very ineffective in terms of cost per unit result. For this reason, the premise that you can “compare methods with regard to outcome” in the absence of associated cost , effort / intensity, and timeline is fundamentally flawed. Failing to clearly acknowledge this reality  is problematic in my judgement, in that it encourages readers (some of whom are not technically savvy) to digest the paper in an effort to glean (or even worse, cherry-pick)  “proof” about what “works” or does not.  It would be much better in the introduction to acknowledge that figuring out “what works” requires integrated consideration of method, effort, and cost, and then positioning the paper as PART of that process, whereby existing publications are reviewed, classified, and summarized, and where deficiencies in the current data gathering effort are highlighted.  Particularly in the Introduction, and less often elsewhere, there are statements that – while generally true – should be qualified in the interest of accuracy. For instance, “most” stray dogs are indeed directly or indirectly dependent in some manner on people, but I don’t know that it is accurate to present that as a universally true statement. TI would read through looking for statements that would be more accurate if qualifiers like “may be”, or “most of”, were included. In determining the “quality” criteria as they apply to study design and analysis, the authors seem to make an assumption that an “idealized” design would be based on a classical hypothesis testing framework, along with classical definitions of power analysis and etc.. While I don’t discount that notion, it is often not practical or efficient to assess management impact in large, messy populations using these approaches, and as a result, many alternative statistical modelling approaches (and study designs) have been developed over the last two decades or more that are increasingly the “gold standards” in ecological and population studies. While I don’t expect that most groups would be able / willing to invoke these approaches, I think it is a disservice for the paper to suggest that a classical approach to quantifying impact is the best that we could do. I’d suggest reviewing this paper (https://royalsocietypublishing.org/doi/full/10.1098/rsbl.2014.0698)  to get a rough sense of what I’m talking about.  All it would take to make the paper more inclusive in this regard is to note that good study designs and associated analyses could be based on classical hypothesis testing, or on more modern process based modeling approaches. The latter have the potential advantages of squeezing more information out of flawed data sets and producing error estimates that are more realistic given a particular data set.  One example of the focus on classical hypothesis testing that I’d call to light is the recommendation of the authors that people trying to estimate management effects identify a control population (line 689). I reality, inclusion of A (singular) control population in no way ensures that cause and effect can  be sorted out in the way suggested in this paragraph, and in fact may increase the chance that spuriously conclude that we’ve obtained that evidence.  There are a multitude of covariates at play that determine whether any local population changes over time, and especially with shorter term data sets, it is difficult to meaningfully distinguish bumps from trends, and to assign causality. So in other words, it is easy to imagine a situation with a single control and single treatment population that rise and fall over a two year period, respectively, and assume (based on a superficially “rigorous” design) that causality has been illustrated, when in fact that are many plausible explanation for why a single population might seem to rise over a two year period while another goes down. Using a sufficiently large sample of control and treatment populations (plural) potential could, of course, help to sort out real effects from noise or from the influence of covariates, but that is likely impractical .  I think this recommendation should be modified to state that obtaining data from control or comparison populations can be valuable if there is sufficient replication, and if information about covariates with a plausible impact on the outcome metric of interest can also be gathered. In the results tables, I’d love to see something more than p values reported as a way to gauge statistical outcomes (if available). To get a sense of how much faith we have in results, it’s awfully helpful to know the analytical method applied, the no. groups (if applicable), sample sizes, error estimates/ CIs/effect sizes, etc. Again, with an eye towards less technical readers who may be hoping to see “evidence” to confirm their pre-existing beliefs, providing only p values without this additional context seems inappropriate. For my taste, the Discussion has too many instances of detailing information that (in the context of this review paper) are actually “results” (i.e. what was reported in a series of individual studies).. I would opt for a slightly shorter Discussion that focuses on the following: Clearly identifies the limitations of the review process, given the information outlined above in point 2. Synthesizes findings for various topics (or notes the lack of coherent information) without going into great detail about the results of individual papers or regurgitating what is / could be in the Results. To be clear, there is already a lot of synthesis in the discussion, I’m just talking about increasing the proportion of synthesis / results. Enhanced discussion of how better data could be collected in the future, based in synthesizing the deficiencies noted in the review process. Note that this is done to a significant extent in section 4.2 already, but it could be improved by including some reference to the topics discussed above in point 4.

Specific Comments

Line45: The sentence starting with “whereas” is not a sentence, but a phrase. Paragraph on line 217: Assessing 2% of papers for ‘rating consistency’ seems like it must be a typo (?).. 2% of 150 papers is just three papers, which is a fairly meaningless sample, and 2% of 20 papers is less than one paper.  Please clarify. Line 342: Just an observation that  a study reporting that population decreased over 9 mo after sterilization and SHELTERING is applied may be a trivial result. Nine mo is likely too fast for even intensive sterilization to have a large effect on population size, whereas the first 9 mo of a sheltering effort will likely pull many dogs off the street and “reduce” the street population over that limited time frame in a very obvious way.  I think it would be fine to provide a little bit of this context in reporting results of papers, which can be amplified in the Discussion section if the context is general. Line 398: An example of a statement that needs qualification, as described above in “General Comments #3). Rewrite “All papers reported that fertility control reduced dog population size” to “All papers reported that fertility control at the intensity modelled reduced population size”.  The idea here (see General Comments #2)  is that you cannot imply that fertility control simply “does” reduce population size except with respect to specific implementation parameters and timelines. Several other statements in this part of the Results section should be similarly modified;  specifically, with “could” words or similar.  All of the modelling results apply only to the extent that their parameters are met (in the best case); their conclusions do not apply under other sets of parameters, and if the papers are good ones, they note that caveat explicitly.

Author Response

We would like to thank all three reviewers for the valuable feedback on the manuscript. We have addressed all comments fully and believe this has vastly improved the manuscript. In addition to addressing these comments, we have included an additional table (table 2), outlining all papers in the final corpus by management method, management intensity, dog population type, study design (including number of groups and replicates). We have also excluded decimal places in text to make the text easier to read. Please find below our response to the reviewers below. 

Reviewer 1: 

Acknowledge in introduction that figuring out “what works” requires integrated consideration of method, effort, and cost and then position the paper as part of that process, whereby existing publications are reviewed, classified and summarized and where deficiencies in the current data gathering effort are highlighted. 

Amended in 1.4 study aims 

Particularly in the introduction, and less often elsewhere, there are statements that – while generally true – should be qualified in the interest of accuracy. For example, most stray dogs are indeed directly or indirectly dependent on some manner on people, but I don’t know that it is accurate to present that as a universally true statement. I would read through looking for statements that would be more accurate if qualifiers like “may be” or “most of” were included.  

Amended in multiple places throughout manuscript. 

In determining the quality criteria as they apply to study design and analysis, the authors seem to make an assumption that an “idealized” design would be based on a classical hypothesis testing framework, along with classical definitions of power analysis etc. It is often not practical or efficient to assess management impact in large, messy populations using these approaches, and as a result, many alternative statistical modelling approaches (and study designs) have been developed over the last two decades or more that are increasingly the “gold standards” in ecological and population studies. While I don’t expect that most groups would be able / willing to invoke these approaches, I think it is a disservice for the paper to suggest that a classical approach to quantifying impact is the best that we could do. I’d suggest reviewing this paper (https://royalsocietypublishing.org/doi/full/10.1098/rsbl.2014.0698)  to get a rough sense of what I’m talking about.  All it would take to make the paper more inclusive in this regard is to note that good study designs and associated analyses could be based on classical hypothesis testing, or on more modern process based modeling approaches. The latter have the potential advantages of squeezing more information out of flawed data sets and producing error estimates that are more realistic given a particular data set. 

Amended in section 4.3 

One example of the focus on classical hypothesis testing that I’d call to light is the recommendation of the authors that people trying to estimate management effects identify a control population (line 689). I reality, inclusion of A (singular) control population in no way ensures that cause and effect can  be sorted out in the way suggested in this paragraph, and in fact may increase the chance that spuriously conclude that we’ve obtained that evidence.  There are a multitude of covariates at play that determine whether any local population changes over time, and especially with shorter term data sets, it is difficult to meaningfully distinguish bumps from trends, and to assign causality. So in other words, it is easy to imagine a situation with a single control and single treatment population that rise and fall over a two year period, respectively, and assume (based on a superficially “rigorous” design) that causality has been illustrated, when in fact that are many plausible explanation for why a single population might seem to rise over a two year period while another goes down. Using a sufficiently large sample of control and treatment populations (plural) potential could, of course, help to sort out real effects from noise or from the influence of covariates, but that is likely impractical. 

Amended in section 4.3 

I think this recommendation should be modified to state that obtaining data from control or comparison populations can be valuable if there is sufficient replication, and if information about covariates with a plausible impact on the outcome metric of interest can also be gathered. 

Modified recommendation in section 4.3 

In the results tables, I’d love to see something more than p values reported as a way to gauge statistical outcomes (if available). To get a sense of how much faith we have in results, it’s awfully helpful to know the analytical method applied, the no. groups (if applicable), sample sizes, error estimates/ CIs/effect sizes, etc. Again, with an eye towards less technical readers who may be hoping to see “evidence” to confirm their pre-existing beliefs, providing only p values without this additional context seems inappropriate: 

Amended in Table 3. 

For my taste, the Discussion has too many instances of detailing information that (in the context of this review paper) are actually “results” (i.e. what was reported in a series of individual studies).. I would opt for a slightly shorter Discussion that focuses on the following: Clearly identifies the limitations of the review process, given the information outlined above in point 2. Synthesizes findings for various topics (or notes the lack of coherent information) without going into great detail about the results of individual papers or regurgitating what is / could be in the Results. To be clear, there is already a lot of synthesis in the discussion, I’m just talking about increasing the proportion of synthesis / results. Enhanced discussion of how better data could be collected in the future, based in synthesizing the deficiencies noted in the review process. Note that this is done to a significant extent in section 4.2 already, but it could be improved by including some reference to the topics discussed above in point 4. 

Restructured discussion and cut out repetition of results. 

Line45: The sentence starting with “whereas” is not a sentence, but a phrase. 

Amended 

Paragraph on line 217: Assessing 2% of papers for ‘rating consistency’ seems like it must be a typo (?).. 2% of 150 papers is just three papers, which is a fairly meaningless sample, and 2% of 20 papers is less than one paper.  Please clarify. 

Amended 

Line 342: Just an observation that  a study reporting that population decreased over 9 mo after sterilization and SHELTERING is applied may be a trivial result. Nine mo is likely too fast for even intensive sterilization to have a large effect on population size, whereas the first 9 mo of a sheltering effort will likely pull many dogs off the street and “reduce” the street population over that limited time frame in a very obvious way.  I think it would be fine to provide a little bit of this context in reporting results of papers, which can be amplified in the Discussion section if the context is general. 

Amended in results 3.4.2 (Lines 417-419). 

Line 398: An example of a statement that needs qualification, as described above in “General Comments #3). Rewrite “All papers reported that fertility control reduced dog population size” to “All papers reported that fertility control at the intensity modelled reduced population size”.  The idea here (see General Comments #2)  is that you cannot imply that fertility control simply “does” reduce population size except with respect to specific implementation parameters and timelines. Several other statements in this part of the Results section should be similarly modified;  specifically, with “could” words or similar.  All of the modelling results apply only to the extent that their parameters are met (in the best case); their conclusions do not apply under other sets of parameters, and if the papers are good ones, they note that caveat explicitly. 

Amended throughout results section for both observational/intervention and modelling studies. 

Reviewer 2 Report

This is a comprehensive systematic review in a very interesting area in which effectiveness is a bit under-researched. Therefore, this study does contribute to current knowledge.

This manuscript is very well written and easy to follow. The method is written in a way that the work can be duplicated and the results and discussion in concisely written. 

My only comments is very minor ones. 

Please keep table format same throughout paper. For example Table 4. has bold in it's descriptive title whereas Table 5. does not. 

Page 25. Please insert a space between line 525 and line 526. 

Author Response

We would like to thank all three reviewers for the valuable feedback on the manuscript. We have addressed all comments fully and believe this has vastly improved the manuscript. In addition to addressing these comments, we have included an additional table (table 2), outlining all papers in the final corpus by management method, management intensity, dog population type, study design (including number of groups and replicates). We have also excluded decimal places in text to make the text easier to read. Please find below our response to the reviewers below. 

Reviewer 2: 

Please keep table format same throughout paper. For example Table 4. has bold in it's descriptive title whereas Table 5. does not. 

Amended in all tables throughout manuscript 

Page 25. Please insert a space between line 525 and line 526.  

Amended 

Reviewer 3 Report

The authors have a done a thorough job of pulling together a topic of potential importance for many countries and disciplines.  My main recommendation is to focus more on the quality of the publications and less on the content details since most of the publications were of relatively low quality based on the authors’ criteria.  I think that there are some sections which are not important and are more of a narrative review. And the recommendations in the discussion are really important! I have specific comments below.

Line 12: there should either be an “s” after stray or another “and”.

Line 45: given the recommendations about population descriptions, please remove “stray” throughout and use owned or unowned as the summary terminology.

Section 1.2: please use a consistent order of topics throughout (this isn’t the same as the tables for example).

I’m not convinced that section 1.3 adds to the systematic review.  This seems like background that would be found or known by anyone involved in this work.  And these references are part of the systematic review itself; many aren’t of high quality either.  If the authors strongly feel this information is needed, it should be shortened and moved to the discussion where it would be integrated into the review summary itself.

Section 1.4: the funding or type of organization driving the studies wasn’t included.  Was there a reason for that? If so, please include in the manuscript. If not, please consider adding some information on that topic and how it might or might not bias the studies.

I would recommend S1 be included in the body of the manuscript. 

Line 236-8: no studies could have only owned (ii or iv) dogs. Please clarify this sentence.

Line 257-60: please edit this sentence for clarity.  I don’t see how you can have most studies coming from  a single country and 15 different countries and in high, middle and low income countries.

Why is figure one important?  If the authors want to keep it please expand on its relevance in the discussion.  I think it could be summarized in a sentence in the results.

Table 1: If I understand the title of this table, the reference follows, not proceeds the number of studies.

Section 3.3:  I think that the quality of the study should be moved before this section AND that Table 2 include the reporting quality indicator score.  Reviewing the findings of these studies without taking into account the quality of the study and therefore the likely reliability and utility of the results decreases the importance of a systematic review.  Table 5 would move to the supplemental material.  Then all of the findings are summarized within the context of the study quality.  That will also mean that the synthesis of results in the text should be more high level, taking into account both the general trends as well as the quality of the papers.  Fewer details on each paper are needed in the text, letting the tables serve that purpose.

Table 2: Reference 70, this seems to say that there was a decrease in normal body condition, but the effect is to increase? Please clarify in the table.  Reference 82: would this be incidence since complications are caused by the surgery?

Lines 502-3:  these are middle income locations so I don’t see how this is supportive of lower income countries having more problems.  Perhaps these are the places where there are both more dogs and more resources for studies?  Please edit.

I would like to see the recommendations moved to line 510.  They are really one of the most important parts of the paper.  And if there are any nuances about why these recommendations are difficult to follow that should be added and acknowledged.

Paragraph starting line 510 to 525: Given that this paragraph is about all of the ways it is too hard to make any summary conclusions, I think that a numbered list of the limitations for that would be shorter and easier to follow. Please edit.

Paragraphs starting line 537 and 556:  These seem repetitive of the results and unnecessarily long. Please shorten and focus on the main points.

Line 584.  If this is a review of all the papers, how is it different from previous work? Please edit for clarity.

Line 603-4: why are these important methods of control?  That argument isn’t clear either in the introduction or here in the discussion. What are the likely mechanisms of impact and are they realistic?

Line 616-8: why is public attitude toward free roaming dogs important and worth more investigation? That hasn’t been made clear in the manuscript.

Line 625: why aren’t these papers part of the review (120 and 121)?

Author Response

We would like to thank all three reviewers for the valuable feedback on the manuscript. We have addressed all comments fully and believe this has vastly improved the manuscript. In addition to addressing these comments, we have included an additional table (table 2), outlining all papers in the final corpus by management method, management intensity, dog population type, study design (including number of groups and replicates). We have also excluded decimal places in text to make the text easier to read. Please find below our response to the reviewers below. 

Reviewer 3: 

Focus more on the quality of the publications and less on the content details, since most of the publications were of relatively low quality based on the authors’ criteria. 

We have amended to include a “limitations” section in the discussion (4.1) and now highlight the tentative conclusions in 4.2. 

Some of the discussion is not important and is more of a narrative review. 

We have amended to exclude a lot of the repetition of results and restructured the discussion. 

Line 12: there should either be an “s” after stray or another “and”. 

Amended 

Line 45: given the recommendations about population descriptions, please remove “stray” throughout and use owned or unowned as the summary terminology. 

Amended 

 Section 1.2: please use a consistent order of topics throughout (this isn’t the same as the tables for example). - Amend 

The order is alphabetical (dog health and welfare; dog demographics; public attitude; public health risk; and risk to wildlife) – I have amended one instance in the text, but all tables are correct (some exclude some topics are no results were found in these areas). 

I’m not convinced that section 1.3 adds to the systematic review.  This seems like background that would be found or known by anyone involved in this work.  And these references are part of the systematic review itself; many aren’t of high quality either.  If the authors strongly feel this information is needed, it should be shortened and moved to the discussion where it would be integrated into the review summary itself. 

I have kept this section, as I think it is important to introduce the reader to the methods likely to be found in the review. Furthermore, the paper is targeted at a general as well as specified audience – as such, this information is needed here. 

Section 1.4: the funding or type of organization driving the studies wasn’t included.  Was there a reason for that? If so, please include in the manuscript. If not, please consider adding some information on that topic and how it might or might not bias the studies.  I would recommend S1 be included in the body of the manuscript. 

Funding and type of organization is now included in supplementary information Table S2. 

Line 236-8: no studies could have only owned (ii or iv) dogs. Please clarify this sentence. 

Amended 

Line 257-60: please edit this sentence for clarity.  I don’t see how you can have most studies coming from  a single country and 15 different countries and in high, middle and low income countries. 

Amended: “Most of the studies were carried out or used data from locations within a single country (87.2%). These were located in 15 different countries across Africa (2.6%), Asia (38.5%), Central America (2.6%), Europe (17.9%), North America (10.3%) and South America (15.4%), in countries that were high income (26.5%), upper-middle income (38.2%), lower-middle income (32.4%) and low income (2.9%). 

Why is figure one important?  If the authors want to keep it please expand on its relevance in the discussion.  I think it could be summarized in a sentence in the results. 

Amended – figure moved to supplementary material 

Table 1: If I understand the title of this table, the reference follows, not proceeds the number of studies. 

Amended 

Section 3.3:  I think that the quality of the study should be moved before this section AND that Table 2 include the reporting quality indicator score.  Reviewing the findings of these studies without taking into account the quality of the study and therefore the likely reliability and utility of the results decreases the importance of a systematic review.  Table 5 would move to the supplemental material.  Then all of the findings are summarized within the context of the study quality.  That will also mean that the synthesis of results in the text should be more high level, taking into account both the general trends as well as the quality of the papers.  Fewer details on each paper are needed in the text, letting the tables serve that purpose. 

Table 5 moved. 

Study quality score included in table 2. 

Table 2: Reference 70, this seems to say that there was a decrease in normal body condition, but the effect is to increase? Please clarify in the table. 

Thank you for spotting this, we have now amended in the table (now table 3). 

Reference 82: would this be incidence since complications are caused by the surgery? 

Amended 

Lines 502-3:  these are middle income locations so I don’t see how this is supportive of lower income countries having more problems.  Perhaps these are the places where there are both more dogs and more resources for studies?  Please edit. 

We’ve now shortened the discussion and taken out this paragraph. 

I would like to see the recommendations moved to line 510.  They are really one of the most important parts of the paper.  And if there are any nuances about why these recommendations are difficult to follow that should be added and acknowledged. 

I have amended the structure of the discussion in line with comments from reviewer 1, this is now structured with: 

Limitations in the review process  Synthesis of findings (brief)  Recommendations for future work  Paragraph starting line 510 to 525: Given that this paragraph is about all of the ways it is too hard to make any summary conclusions, I think that a numbered list of the limitations for that would be shorter and easier to follow. Please edit. 

Amended to include numbered list (Lines 602-638) 

Paragraphs starting line 537 and 556:  These seem repetitive of the results and unnecessarily long. Please shorten and focus on the main points. 

Amended to remove repetition of results. 

Line 584.  If this is a review of all the papers, how is it different from previous work? Please edit for clarity. 

I’m not sure if this refers to the reference, or the text. 

Line 603-4: why are these important methods of control?  That argument isn’t clear either in the introduction or here in the discussion. What are the likely mechanisms of impact and are they realistic? 

Excluded from discussion 

Line 616-8: why is public attitude toward free roaming dogs important and worth more investigation? That hasn’t been made clear in the manuscript. 

Amended 

 Line 625: why aren’t these papers part of the review (120 and 121)? 

120: Cleaveland et al (2006). Canine vaccination – providing broader benefits for disease control. This paper is a review article and does not meet the eligibility requirement: (i) primary research source. 

121: Hampson et al (2009). Transmission dynamics and prospects for the elimination of canine rabies.  This paper was excluded at stage three (review of full text) on the basis of relevance to the specific research question. Hampson et al is investigating the impact of vaccination rather than dog population management.